

# The effects of tillage methods on soil aggregation and crop yields in a wheat-corn rotation under semi-arid conditions

Hossein Tabiehzad[1], Gokhan Cayci[2], Kiarash Afshar Pour Rezaeieh[3*]

[1]Department of soil and water, Agricultural Research Center of West Azerbaijan, Urmia, Iran
[2]Department of Soil Science and Plant Nutrition, Faculty of Agriculture, Ankara University, Ankara, Turkey
[3]Department of Field Crops, Agricultural Faculty, Igdir University, Igdir, Turkey
Corresponding author: febspark7@gmail.com

## Abstract

In this study, the effects of different tillage methods under wheat-corn two-course rotation system on the some soil aggregation properties and yields were investigated. Experiment was laid out in a split plot design with three replications during four crop years. Subsoiler, moldboard, sweep and chisel as main plots and rotary tiller and disc harrow as sub-plots have been used in this study. The results showed that tillage methods were significant at ($P<0.01$) as regards crop yields, and the highest yields as 6249 and 11720 kg/ha for wheat and 9891 and 73080 kg/ha for corn grain and biomass were produced in subsoiler treatment, respectively. Subsoiler+rotarytiller treatment was significant at ($P<0.05$) with 2.063 mm as to mean weight diameter (MWD) value. The subsoiler and chisel were statistically in the same group with regard of water stable aggregates (WSA) value, and it was significant at ($P<0.05$) with 67,83%. Bulk density, total porosity and air porosity values were significant at ($P<0.01$), and 1.38 grcm$^{-3}$, 51.2% and 12.5% values were determined in rotary tiller application, respectively. Field capacity (FC) and permanent wilting point (PWP) were significant at ($P<0.05$) and ($P<0.01$) with 31.89% and 17.21% values in the chisel treatment, respectively. Crop yields and positive effects on the physical properties were considered subsoiler+rotary tiller treatment was the most successful, and it was followed by chisel+rotary tiller treatment according to four-year study results.

**Key words:** Soil quality, subsoiler, chiesel, soil structure





## 1. Introduction

The soil as a natural resource for the future of mankind must necessarily be managed in a sustainable manner. Thus, to preserve agriculture for future generations, developing production systems that conserve and enhance soil quality is fundamental (Doran and Zeiss, 2000). The retention of water and soil conservation are important in arid and semi-arid regions. Additionally, maintaining of organic matter is so difficult because of high temperature at these regions (Laegried et al. 1999). Gumus and Seker (2015) have pointed to the importance of organic matter in sustainable soil productivity in semi-arid areas where organic matter is low. It is well known organic carbon is more accumulated in the surface part of the soil. Sufficient level of organic matter in the top soil improves the physical, chemical, and biological features and thereby qualities of soils (Sojka and Upchurch 1999). On the other hand, studies on soil tillage show that the most accumulation of organic matter has been found in direct sowing, minimum and conventional tillage methods, respectively (Anonymous 2002).

The amount of organic matter in soil and soil aggregate stability is low in arid and semi-arid regions. Due to irregularities in rainfall with regard to time and intensity in arid and semi-arid regions, soil erosion and soil loss increases and aggregate stability decreases. Moreover, excessive or incorrect tillage reduce the soil characteristics and quality (Chenu et al. 2000; Marinari et al. 2000). Alakkuku et al. (2003) reported that subsoil compaction due to increase of field traffic is a serious problem, because the effects are long-lasting and difficult to correct.

Tillage systems are basically evaluated in two categories: conventional tillage systems and conservational tillage systems. Conservational tillage covers the methods such as; minimum tillage, zero tillage, mulch tillage, ridge tillage and line tillage (Holland, 2004). It is mentioned that conservational tillage can reduce the yield in first years of implementation but it offers more protection against soil degradation and more improvement in quality of soil in long term (Lampurlanes et al. 2001). Primary tillage implements such as moldboard, turning the soil upside down, and excessive tillage practices are matter of concern in conventional tillage. Plant residues decompose in this method very quickly. Besides, it also leads to soil erosion and degradation and considerable amount of soil carbon exhaust in the form of carbon dioxide gas (Glanz 1995).





Conservational tillage does not cover the moldboard methods but consists minimum tillage,
direct sowing, and zero tillage as limited tillage. 15 up to 30 percent of plant residues
remain on the surface part of the soil at this system and therefore, cultivator or herbicide is
used to control the weeds (Gajri et al. 2002).
Tillage is one of the most important components of crop production that farmers have used
to it more on the basis of their experiences. Publications related to cultivation and tillage,
has been focused more on product yield than of changes appear in soil properties through
the various tillage methods. Tillage practices today affects soil fertility and environmental
quality. It will impact some restrictive soil properties, improve its properties and increase in
the crop yield if practices reasonably and consciously (Lal 1991). Tillage affects soil
fertility in short term and quality of soil in long term (Gajri et al. 2002). In spite of
Bhattacharya et al. (2006), who believed that the soil tillage methods, in addition to improve
of physical properties and content of organic matter and soil characteristics, leads to
changes in soil fertility, Melero et al. (2011) argued that the effects of soil tillage methods
on physical properties varies and is not guaranteed. Similarly, Strudly et al. (2008) indicated
soil tillage studies can display changes depending on the experimental designs and trial
locations. Srivastava and Meyer (1998) reported that soil tillage systems could have
advantages and disadvantages in different situations, but there is no an ideal single system
in all soil, climate and crop conditions.
No-till farming practices are increasing in recent years, although, it is required to the
preparation of the seed bed in arid and semi- arid regions of mechanical operations. Today,
instead of excess tillage and conventional tillage methods, in some areas farmers have
begun to use protected or reduced tillage methods. These methods are combined with
reduced tillage practices using tools and equipment, and they are preferred especially for
strategic products such as wheat and corn.
In Urmia ($37°33′19″$N $45°04′21″$E), located in the northern west Iran, farmers use
conventional tillage methods as the first plow tillage and disk harrow as the second to
prepare the soil. The common rotation systems in the region are as wheat-sunflower, wheat-
sugar beet and wheat-corn. The aim of this study was to determine the effects of tillage
methods on wheat-corn double rotation system, soil aggregation characteristics and yield of
wheat and corn as well.



**2. Materials and Methods**
**2.1 Material**
**2.1.1 Trial Location**
The trial was conducted at Saaetloo Agricultural Research Station (37 ° 43 ' 31 " N and 45 °
01' 59 ") located 20 kilometers north of Urumia, Iran.
**2.1.2   Soil and climate characteristics of the experimental field**
Some soil properties of the experimental field and some climate characteristics are given in
Table 1 and 2. The field is flat with very low slope. Top soil texture (0-60cm) is silty clay
loam and getting heavier in the deeper depths as silty clay. According to Soil Survey Staff
(2006), soil is classified as fine, mixed, super active, and it is mesic Typic Haploxerepts.
**2.1.3  The plants**
" Zerrin" wheat variety and Yugoslavia silage corn (SC704) were used as plant materials.

Table 1. Soil properties of the experimental field

| Depth (cm) | Clay (%) | Silt (%) | Sand (%) | Saturation (%) | N (%) | P (ppm) | K (ppm) | Organic carbon (%) | CaCO$_3$ (%) | pH | EC (dS/m) |
|---|---|---|---|---|---|---|---|---|---|---|---|
| 0-30 | 42 | 47 | 11 | 50 | 0.095 | 4.51 | 396 | 0,95 | 18 | 7.60 | 1.49 |


Table 2. Means of the maximum temperature (°C) and average total precipitation (mm) during
2002-2012.

| Month / Year | January | February | March | April | May | June | July | August | September | October | November | December | Average |
|---|---|---|---|---|---|---|---|---|---|---|---|---|---|
| Temperature (°C) | -6.43 | -4.45 | -1.78 | 3.21 | 7.37 | 110 | 5.22 | 15.64 | 11.8 | 7.45 | 2.71 | 0 | 5.19 |
| Precipitation (mm) | 19.9 | 30.6 | 39.6 | 48.9 | 39.,7 | 11.8 | 4.42 | 3.49 | 6.76 | 29.7 | 45.8 | 0 | 297.8 |


**2.2 Method**
**2.2.1 Treatments**
The experiment was carried out in a split-plot block design with three replications. Four
tillage implementations were as the main plots and the other two tillage methods as the sub
plots under wheat-corn rotation system. This rotation system is one of the most preferred
system of farmers in the region. Wheat cultivated at the range of 160-180 kg/ha. After the
wheat harvest, all tillage methods were applied to the plots and silage corn was planted in




the spring. 135, 200, 150 and 130 kg/ha urea and 130, 135, 155 and 115 kg/ha triple super
phosphate were applied to the wheat in 2009-2010, 2010-2011, 2011-2012 and 2012-2013
growing seasons, respectively. On the other hand, 350, 320 and 300 kg/ha urea and 150,
160 and 175 kg/ha triple super phosphate were applied to the corn in 2010-2011, 2011-
2012 and 2012-2013 growing seasons, respectively.
**2.2.2 Soil analyses**
Soil samples were taken from the depth of 0-15cm after the wheat harvest in 2013. Eight
samples were taken from each plot and mixed each other and the soil sample obtained from
this mixture was used for analysis. Soil texture was determined by hydrometer method
described by Bouyoucos (1951), textural classes were determined using the texture triangle
specified by Soil Survey Division Staff (1993). Saturating the soil samples with water, soil
acidity of the soil-water extract 1/2.5 (w/w) by the pH meter, the electrical conductivity of
the saturation extract from a soil-water paste by EC meter and bulk density at undisturbed
soil samples were determined according to U.S. Salinity Laboratory Staff (1954). The
moisture contents of undisturbed samples at field capacity and wilting point were
determined using pressure plate (Cassel and Nielsen 1986). Macro and micro pore amounts
in undisturbed samples were measured using by porous ceramic plates creating a negative
pressure of 50 cm (Romano et al. 2002). Saturated hydraulic conductivity of undisturbed
samples was measured using constant head permeameter (Klute and Dirksen 1986).
Aggregate stability was determined as reported by Kemper and Rosenau (1986). Mean
weight diameter was calculated as indicated by Hillel (1980) the diameter size distribution
considering the dry aggregates. Organic carbon was measured as indicated by Nelson
(1982). Calcium carbonate content was determined by calsimeter method according to
Nelson (1982). Total nitrogen was analyzed as reported by Bremner (1982) applying the
micro-kjeldahl method**.** Available phosphorus was determined as indicated Olsen et al.
(1954). Available potassium was measured as noted by Jackson (1958).
**2.2.3 Statistical analysis**
Statistical analysis was performed using MSTAT-C program.
**3. Results and Discussion**
**3.1 Grain and biomass yield of wheat and corn**
Significant effects of tillage methods on grain and biomass of yield of wheat and corn and
aggregation properties of the soil were determined. Treatments on wheat grain yield and

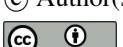


biomass production has been identified distinct and the four year application results of the
variance analysis of the main-plots and sub-plots was found to be significant at the (p<0.01)
level (Table 3). The highest wheat grain and biomass yield (6249 kg/ha and 11720 kg/ha,
respectively) was identified in subsoiler application, while the lowest amounts in the same
order was determined as 4777 kg/ha and 9770 kg/ha in the mouldboard plough practice.
According to the results of three-year variance analysis, corn grain and biomass yield
productions at the sub-plots and main plots were significant at the (p<0.01) level. Maximum
corn grain yield and biomass production were recorded (9891 kg/ha and 73 080 kg/ha,
respectively) subsoiler treatment, while the lowest amounts in the same order were again
determined as 8176 kg/ha and 57350 kg/ha in the mouldboard plough practice. (Table 3).
Subsoiler has been identified more effective than the other tillage methods both wheat and
corn. Deep tillage practices in semi-arid regions of India has been proved to be effective in
corn and sunflower cultivation by Arora (1991) and Gajri et al. (1997). In current study,
chisel practice took the second place in increasing both grain and biomass production of
wheat. In corn growing, subsoiler and chisel practices were in the same group, statistically.

Table 3. Effect of tillage methods on wheat and corn grain-biomass yield (kg/ha)

| Tillage Methods | Wheat biomass | | Wheat grain | | Corn biomass | | Corn grain | |
|---|---|---|---|---|---|---|---|---|
| Subsoiler | 11720 | **a** | 6249 | **a** | 73080 | **a** | 9891 | **a** |
| Sweep | 10400 | **b** | 5593 | **b** | 69370 | **a** | 9023 | **ab** |
| Moldboard | 10330 | **bc** | 5045 | **c** | 57010 | **b** | 8853 | **b** |
| Chisel | 9770 | **c** | 4777 | **c** | 57350 | **b** | 8176 | **b** |
| Probability   P< 0.01 | LSD: 599,7 | | LSD: 379,3 | | LSD: 6178 | | LSD: 970 | |
| Tillage Methods | Wheat biomass | | Wheat grain | | Corn biomass | | Corn grain | |
| Rotary tiller | 10794 | **a** | 5601 | **a** | 68882 | **a** | 9442 | **a** |
| Disc harrow | 10317 | **b** | 5230 | **b** | 59525 | **b** | 8528 | **b** |
| Probability % | P<0.01 | | P<0.01 | | P<0.01 | | P<0.01 | |

Statistically significant difference between the means is shown in separate letters



These data are consistent with results of previous studies including Diaz- Zorita and Grasso,
2000). According to Oussibl and Crookston (1987), deep subsoiler practice resulted in a
54% yield increase by providing better wheat root growth and crop development, and it
causes a decrease in bulk density. Hajabbasi and Hemmat (2000) determined 7264 and 6815
kg/ha wheat grain yields under conventional and non-inversion tillage systems, respectively
as the four-year yield average. De vita et al. (2007) pointed to the amount of rainfall in
wheat yield and found that the no- tillage system should be preferred on continuous durum
wheat growing areas with a lower rainfall of 300mm, whereas more rainy areas
conventional tillage increased the wheat yields. Su et al. (2007) reported the winter wheat
yields were higher on no-tillage with mulching and subsoil tillage with mulching treatments
compared with conventional tillage, and proposed no-tillage and subsoil tillage systems
were the optimum tillage systems to increase yield, water storage and water use efficiency.
Root length density of corn was found to be effected by soil tillage systems, and the order
was moaldboard plow>chisel plow>no-till in the upper layers of soil (Mosaddaghi et al.
2008). On the other hand, Godwin (1990) stated that every year application of  subsoiler is
not appropriate because it is an expensive process, but farmers may consider subsoiler
tillage method to tillage rotation when needed.
In rotary tiller, wheat grain and biomass yields of the sub-plots were 5601 kg/ha and 10794
kg/ha, respectively (Table 3). Compared to disc harrows, rotary tiller implementation
resulted in an increase of 417 kg/ha and 371 kg/ha wheat grain and biomass, respectively.
In addition, rotary tiller method led to a production of 9442 kg/ha grain and 68882 kg/ha
biomass in corn. Compared to disc harrow, rotary tiller produced more corn grain and
biomass of 1114 kg/ha and 9357kg/ha, respectively (Table 3). The outputs from current
study confirms previous findings (Ozpınar and Cay 2005).
**3.2 Mean weight diameter of aggregates and hydraulic conductivity**
Mean weight diameter (MWD) results are given in Figures 1 and 2. Compared to other soil
tillage methods, MWD in the main plots was higher with 1.907 mm in subsoiler tillage
application (Figure 1). Considering main and sub plot interaction, subsoiler + rotary tiller
resulted in 2.063 mm MWD, while the lowest value was devoted to chisel + disc harrow
method (Figure 2). Bybordi (1990) stated that high MWD value implies high soil aggregate
stability. Follette (2001) mentioned that tillage methods have significant effect on MWD.

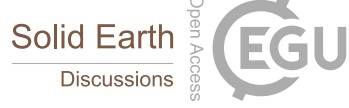



Jaiyeoba (2003) reported a decrease of the stability of the aggregates because of
conventional tillage practices. Filho et al. (2002) determined lower MWD in conventional
tillage rather than zero tillage.

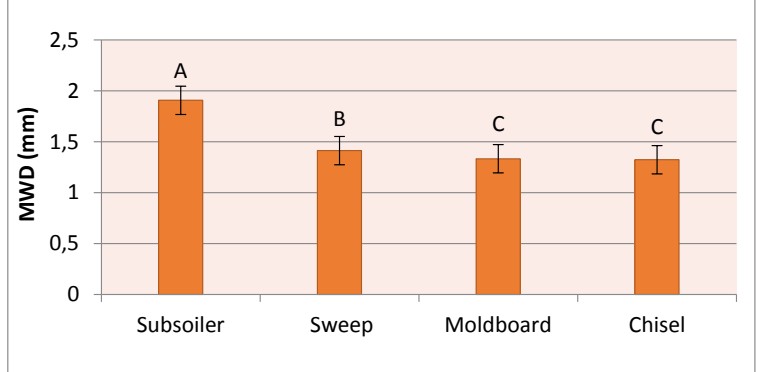


Figure 1. Effect of tillage methods on the mean weight diameter (mm)
The difference between the averages shown in separate letters (P <0.01)
The error bars show SE values.

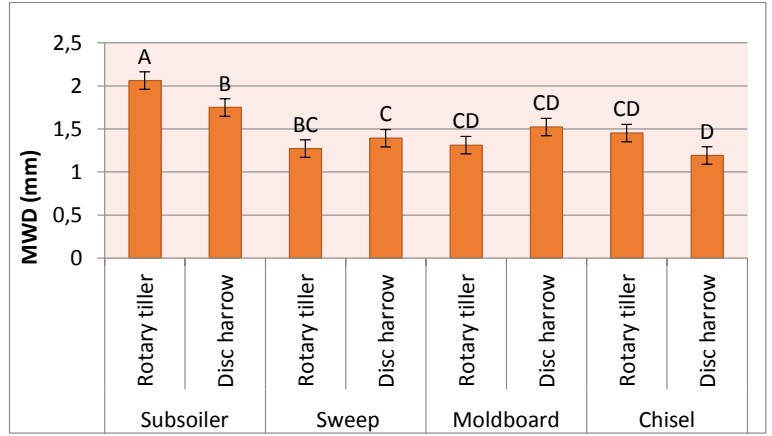


Figure 2. Effects of tillage methods on the mean weight diameter (mm).
The difference between the averages shown in separate letters (P <0.05)
The error bars show SE values.

Hydraulic conductivity is closely related to water movement in the soil and tillage practices
can influence on this feature. Variance analysis of tillage methods on hydraulic conductivity
is given in Table 4.






Table 4. Effect of tillage methods on hydraulic conductivity (cm/h)

| Tillage Method | Subsoiler | | Sweep | | Moldboard | | Chisel | | Sub plot means | |
|---|---|---|---|---|---|---|---|---|---|---|
| Rotary tiller | 1.57 | a | 0.32 | c | 0.26 | c | 0.91 | bc | 1.57 | a |
| Disc harrow | 1.04 | b | 0.29 | c | 0 .23 | c | 0.52 | c | 1.04 | b |
| Main plots  means | 1.305 | a | 0.30 | c | 0.245 | c | 0.715 | b | | |

Statistically significant difference between the means is shown in separate letters.
Main plot: $P< 0.01$, Sub plot: $P<0.05$, Main plot - sub-plot interaction: $P<0.05$

Subsoiler treatment of 1.305 cm/h was found to have more hydraulic conductivity than the
other tillage methods (Table 4), followed by chisel. The lowest value was determined in
moldboard plow measured as 0.245 cm/h. Subsoiler + rotary tiller treatment with 1.57 cm/h
showed the highest value. Kahlon et al. (2013) reported that tillage systems can change the
number, shape, continuity and size distribution of pore network. According to Germann et
al. (1984) in the soil profile, water distribution and infiltration of conventional tillage
system is as twice as zero tillage. Ahuja et al. (1989) reported that large voids are
responsible for the effective porosity in soil, so that hydraulic conductivity and infiltration
amounts in soil are affected by soil tillage methods. On the other hand, Osunbitan et al.
(2005) indicated that the disturbance of continuity of macro pores under the conventional
tillage is the most important factor for saturated hydraulic conductivity in the soils. It was
noted that disc harrow, even in low levels, can cause compaction in the fields where there
were no plant residues and non-mulching materials (Davies et al. 1999; Ghuman and Sur

2001).

**3.3 Bulk density, air porosity and total porosity**
Effects of tillage methods on bulk density, porosity and total porosity are given in Table 5.
Rotary tiller application has led to 1,304gr cm$^{-3}$ bulk density comparing mean values. While
in disc harrow application bulk density was found to be 1.394 g cm$^{-3}$.
Table 5. The effects of tillage methods on bulk density, air porosity and total porosity

| Tillage Method | Bulk density (gr cm$^{-3}$) | | Air porosity (%) | | Total porosity (%) | |
|---|---|---|---|---|---|---|
| Rotary tiller | 1. 304 | a | 12.05 | a | 51 .26 | a |
| Disc harrow | 1.394 | b | 10.60 | b | 49.48 | b |
| Probability % | $P<0.0 1$ | | $P<0.05$ | | $P<0.01$ | |

Statistically significant difference between the means is shown in separate letters.
Soil tillage was found to decrease bulk density and hydraulic conductivity of the soil (Meek
et al., 1992). Ozpınar and Cay (2005) reported effects of moldboard plow, disc harrow and



rotary tiller methods on soil properties and the wheat yield. Bulk density values for rotary
tiller, moldboard and disc harrow applications were determined as 1.20, 1.34 and 1.24 gr
cm$^{-3}$ for 0-10cm depth, 1.26, 1.29 and 1.21 gr cm$^{-3}$ for 10-20cm depth and 1.30, 1.27 and
1.40 gr cm$^{-3}$ for 20-30cm depth, respectively. Pierce and Burpee (1995) reported increase in
crop yield and total porosity value, while decrease in bulk density as a result of subsoiler
application. Many authors indicated decrease in yield and increase the bulk density by
increasing farm traffic (Zhang et al. 2006). While Lal (1999) identified chisel or moldboard
application had no effect on bulk density of fluffy soil. Baldev Singh and Malhi (2006)
reported soil bulk density in rotary tiller as 0.99 gr cm$^{-3}$ and 1.41 gr cm$^{-3}$ under straw
removed and straw retained practices, respectively. Roseberg and McCoy (1992) reported
increased total porosity and decreased number of effective pore and continuity in the
conventional tillage method.
The highest soil porosity with %51,2 was determined in rotary tiller application when the
averages of the tillage practices compared with each other. While the lowest amount was
identified in disc harrow with %49,4 (Table 5). Godwin (1990) reported soil aeration is
relevant to total pore amount and the percentage of macro pores, and identified air porosity
of 10% and above enough for many crops. Same author noted that fragmentation of soil
aggregates and breaking of the pore continuity leads to reduced air porosity by increasing
the retention of soil water. Abu-hamdeh (2003) stated that increase in wheat yield by
increasing of soil aeration in compacted soils. Shiptalo and Protze (1987) investigated the
effects of tillage on soil morphology and porosity and found the amount of macro pores in
Ap horizons of no- tillage is about half of that found in the conventional tillage. Xu and
Mermoud (2001) have indicated that subsoiling tillage increases the amount of larger pores
(>50µm diameter) but reduces the amount of smaller pores (<10 µm diameter). Increments
in larger pores in subsoiled soil resulted in increases in hydraulic conductivity and
infiltration rate compared to no-tillage soil.
**3.4 Water stable aggregates**
Effects of soil tillage methods on water stable aggregates are shown in Figure 3. Subsoiler
and chisel treatments were in the same statistical group when comparing mean values. The
maximum amount of water stable aggregate was found in the subsoiler application with
67.83 %. The least water stable aggregate amount was determined in the moldboard plow
with 47.67 %.






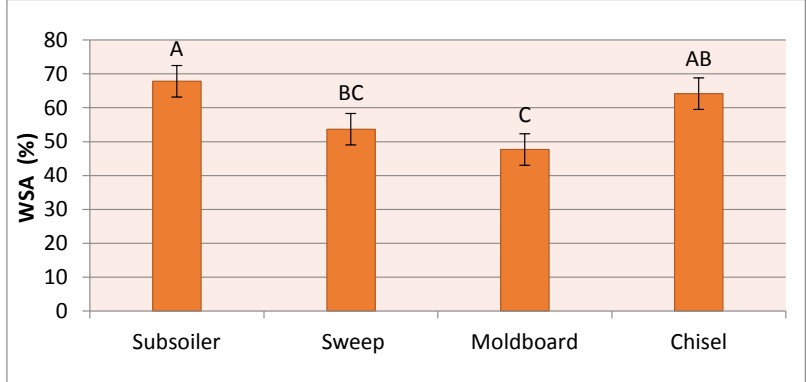


Figure 3. Effects of tillage methods on water stable aggregates (%)
Statistically significant difference between the means is shown in separate letters (P<0.05)
The error bars show SE values.

In the soil management, persistency and stability of soil aggregation is associated with the
size of aggregates (Traore et al. 2000; Whalen and Chang 2002). Nyamangar et al. (1999)
indicated that there is need to root secretions in the soil for aggregates to be increased.
Martens (2000) reported increases of water stable aggregates as a result of corn crop
residues, and suggested less soil inverting methods to increase soil aggregation. Shaver et
al. (2002) noted that no till cropping in wheat-corn rotation returned more crop residue,
decreased bulk density, increased porosity and improved soil aggregation compared to
wheat-fallow. Bronik and Lal (2005) noticed the effectiveness of organic matter and
decomposition degree on the stability of aggregates. On the other hand, Abiven et al (2008)
determined the correlation between decomposition characters of crop residues and soil
aggregates. Meanwhile Shaver et al. (2002) and McVay (2006) determined increase of
macro aggregates and total porosity due to high aggregate stability which in turn causes in
high infiltration and water use efficiency. Kasper et al. (2009) determined the amounts of
water stable aggregates under conventional and reduced tillage treatments as 18,2 % and
18,9%, respectively, whereas it was found as 37,6% at minimum tillage practice. Besides,
authors noted conventional tillage interfere more natural soil properties than reduced and
minimum tillage.
**3.5 Field capacity and wilting point**
The effect of the tillage methods on field capacity is given in Figure 4. Comparing field
capacity and soil tillage methods with each other revealed chisel and subsoiler methods in




309 the same statistical group, devoting 31.89% and 31.90%, respectively. The lowest field

310 capacity values were obtained from the sweep and moldboard applications. When compared

311 the amounts of water retained at field capacity subsoiler, chisel and moldboard treatments

312 were found in the same statistical group, while lower amount was determined in the sweep

313 plow.

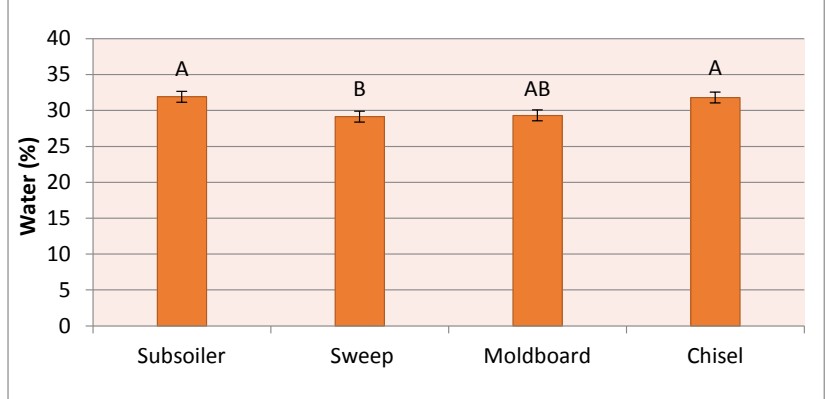

316 Figure 4. Effects of tillage methods on field capacity (%)
317 Statistically significant difference between the means is shown in separate letters (P<0.05).
318 The error bars show SE values.

320 Boone (1988) noted that tillage practices are effective on the amount of water both surface

321 and below the soil. Accordingly, the changes due to the field traffic, affects the amount of

322 porosity, number and continuity of the pores and the hydraulic properties of the soil.

323 Logsdon et al. (1990) reported breaking of the continuity of pores in the soil in moldboard

324 plow application, while the continuity was maintained in the ridge planting method. In a

325 study carried out by Mahboubi et al. (1993), between soil tillage practices in terms of

326 available water content, the order was zero tillage> chisel plow>moldboard plow. On the

327 other hand, Vepraskas (1988) reported that the increase in bulk density causes an increase in

328 penetration resistance and a decrease in the available water amount.

329 Effects of soil tillage methods on wilting point are shown in Figure 5. Comparing methods

330 of tillage revealed chisel plow, moldbord plow and subsoiler plow in the same statistical

331 group, while the highest moisture content (17.21%) was determined at the wilting point in

332 the chisel plow, and lowest value (15.78%) was found in sweep plow (Figure 5).




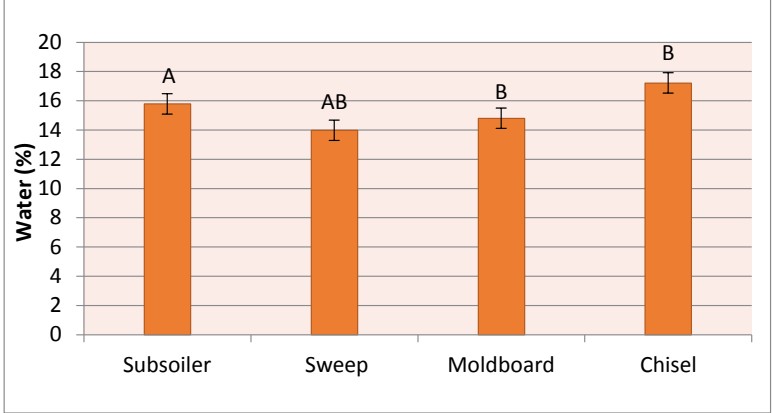


Figure 5. Effects of tillage methods on wilting point (%)
Statistically significant difference between the means is shown in separate letters (P<0.05).

Bescanza et al. (2006) indicated the amount of available water in the tilled soils is related to
the reorganization of pores and reported no-tillage and reduced tillage had higher soil water
contents than the moldboard plowing at matric potentials of 0 to -1500 kPa according to the
driest year of the five-year study period. Su et al. (2007) investigated the combined effects
of tillage and mulching on soil water. They determined the soil water storage quantity was
25 and 24mm higher in the mulching treatments of no-tillage and subsoil tillage,
respectively, than conventional tillage and reduced tillage treatments according to six–year
study results. Mahboubi et al. (1993) reported that available water holding capacity was in
the order of no-till>chisel plowing>moldboard plowing according to long-term tillage
experiments.
**4. Conclusions**
Considering wheat and corn grain and biomass yield, the most appropriate tillage method
was identified as subsoiler + rotary tiller followed by chisel + rotary tiller application.
In compact soils suffering from intensive agriculture practices, subsoiler application brough
about formation of cavities by breaking the lower layers, loosening of the soil and
increasing air, water and heat movements. Thus, the movement of water in the soil would be
facilitated and plant root depth in terms of better physical conditions would be provided.
On the other hand, subsoiler accompanying rotary tiller application increases wheat and
corn seeds contact with the soil, provides the proper seed bed preparation and improves soil
aeration. The study showed that the yield and soil properties were superior to rotary tiller
considering the disc harrow. According to the research results, subsoiler and chisel were in





the same group statistically in the point of most of the soil properties. The highest hydraulic conductivity, the mean weight diameter, amount of water-stable aggregate and field capacity values were found in the subsoiler. On the other hand, the highest total porosity and air porosity values were determined in the rotary tiller application. As a result of this research, considering either wheat or corn grain or biomass yields along with studied soil properties, subsoiler was the most suitable tillage method accompanying rotary tiller. In terms of efficiency and positive impact on soil properties, chisel application after subsoiler has been found to be applicable and promising. Considering the high energy costs of the subsoiler application, in wheat - corn rotation system a subsoiler + rotary tiller for every three or four years is preferable. While in other years, chisel + rotary tiller application may be suggested as the result of this experiment for practitioners.

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
