# Peer review of "The effects of tillage methods on soil aggregation and crop yields in a"

_Solid Earth, 2017_

## Referee Comment (RC1) · Anonymous Referee #1 · 28 Mar 2017

General comment:

The authors present a study about the influence of tillage methods on soil physical properties and crop yields in a wheat corn rotation in Iran which goes to subjects the journals publications scope. The paper is interesting, but it is not performed to be well understood. The paper have some interest in given subject, but shows several weaknesses which should be changed and discussed. Therefore I suggest reject & re-submission grade in order to improve quality.

Mayor points: The paper is "wordy", but on the other hand not focused in important findings (e.g. authors do not connect the yields and biomass with physical soils state at each treatment, neither explain their results). The reader gets lost in all the words.

Condense the manuscript. Remove all unnecessary statements and data. Please, avoid repetition and make sure that all statements are credible. The English is not understandable in many places. Also, there is lack of hypothesis in this study and novelty of the study can be explained. Effects of tillage management on plant, soil and environment certainly increase knowledge of how to build more sustainable agricultural systems in Iran. Nevertheless, this subject is extensively studied around the Earth so I wonder how attractive is this subject on international level? The main problem with the paper is that the authors did not provide description of treatments, tool that was used and depth of interventions. This makes the manuscript impossible to evaluate. Introduction section should be rewritten, while recent literature should be used to describe the actual problems. Paper can be better organised – I add some suggestions into text. Furthermore, your data deserves better discussion. Discussion is poor and does not explain neither supports the study results. The authors should say why they have it and not present only the results of the others. More details about sampling procedure are needed.

Specific comments:

Abstract Abstract should be rewritten. Please rewrite according next guidelines: Include more information to encourage readers to continue. Many only read title and Abstract. State where you perform the research. State objectives of research, and show whether data meet objectives. State hypotheses tested, and show whether data support hypotheses. Include international classification of soils. Include more data to support statements, and give better summary of the manuscript. Line 14: define "two-course" Line 15: Define "some aggregation properties"? Is better to use "physical properties" or state each property. Line 16: delete "crop" Line 19: Is better to show the results in tonnes per ha. Please change in whole manuscript. Line 21-22. Unclear, please rewrite. Line 24-28. It is hardly understandable. Please ensure English improvement and editing from English speaker.

Introduction Please organize the introduction of the paper better. 1: General aspects

of tillage impacts on the agro-ecosystems 2: Background about tillage impacts on soil properties that you are studying in this paper. 3: Impacts of tillage and implications trough soil properties (that you study) on crops that you study (generally and in similar agro-ecological conditions). 4: Justify why your study is needed and the novelty of your work. 5: Hypothesis that you want to test in your work and aim.

Line 38: write "agricultural" instead "production" Line 39: write "Soil and water conservation are important..." Line 43: Please, use one term, not many of them – e.g. soil organic carbon – soil organic matter Line 46-48: There is plenty of research available to support this statement. Add newer literature. Line 49-50: Avoid repetitions Line 50-51: add "distribution" below "rainfall" and "rainfall events" after "intensity" Line 52: Please avoid generally writing. State which soil characteristics. Line 53-55. Statement is not true. It is not difficult to make intervention (subsoiling) but is expensive. Line 67: unify terms: zero tillage – no tillage – direct sowing. Line 67-69: State in which method? Line 71: Cultivation is tillage. Please delete "cultivation" Line 73-76: Avoid general statements. Instead is better to specify which parameter will be changed and how in short and long term. Support each mentioned parameter with literature source. Line 87: State the difference between excessive tillage and conventional tillage Line 88: write "conservation" instead "protected or reduced"

End of introduction: state hypothesis and explain novelty.

Introduction: Study that deals with comparison of subsoiling (ripping) with conventional tillage is missing and their effect on soil physical properties and yield.

Materials and Methods Better explanation of experimental design is needed. State the area of each treatment. State the area of sub-treatment. Explanation and description of tillage procedures, tools and depth of tillage is missing. Essential for paper. Line 102: Urumia or Urmia? Line 105: use "terrain" instead "field". Delete "with very low slope" Table 1: use mg/kg instead ppm Table 2: Check the temperature in June. Remove the comma in May Line 121: which year? Line 122: which year? Soil analyses: Sampling

strategy is poorly written. How many disturbed samples per plot were collected? How many undisturbed samples per plot were collected? Same for subplots. How many disturbed and undisturbed samples in total? For undisturbed samples: from which depth? Or depths? Determination procedure for aggregate stability needs some more details. . .

Statistical analysis should be described well.

Results and discussion: This section need to be reordered: 1) bulk density, air filled porosity and hydraulic conductivity, 2) MWD and WSA, 3) field capacity and wilting point, 4) biomass and grain yields. Try to present results in order presented in tables and figures. Highlight most important finding, and state the significant differences between treatments rather than present relative numbers. Discussion is poorly written and does not support, neither explain given differences between treatments.

Line 158 and 163: it is not true. Line 165: Deep tillage means nothing. Provide specific information about tillage operation. Line 165-166: compared to what treatment? Line 168: This statement is not supported with results

Table 3: state moisture content in grain and biomass. Does numbers represent average of all four seasons? Provide main plot x subplot interaction. Same for each year.

Line 173-196: Suggest to authors try to explain their results rather than report other findings. Find and compare your results with study results performed on similar textured soils and climatic conditions.

Figures: remove horizontal lines. Unify x –axis and put white background. Table 4: unify digits, remove blank under "molboard" row

Line 245: use g per cm3 not gr per cm3. Please make adjustment in whole manuscript. Table 5: change letters in bulk density Line 253: put space after 10 Line 257: Use term "vehicular" instead "farm" Line 258: define "fluffy" Line 258-260: please delete. Does not refers to present subject Line 261: define "effective pore" Line 263 and 265: put %

behind numbers Line 293-296: Does this study deals with crop rotation effect? Please delete this Line 297-299: What is procedure with residues in this experiment? Add this information in M&M section Line 301-303: When you compare results with others please provide information about tool that were used and depth of intervention instead using terms "reduced" and "minimum". Otherwise is meaningless comparison.

Conclusions Line 349-350: Authors presents only single factor effect. If you want a draw a conclusion similar to written than you should test an interaction effect. This statement is not confirmed. Line 351-353: did you study subsoil layers also? Or just topsoil? Please delete this, it is not investigated. Make conclusions from your results. Line 366-369: Effect of subsoiling on silty clay soils do not last longer than season or two. Please adjust conclusion like: …... "subsoiling is suggested in years when crops with demands for deeper rooting occurs in crop rotation."

Literature There are only two (2) sources in last 5 years. I suggest refreshing the paper with newer literature.

---

## Referee Comment (RC2) · Anonymous Referee #2 · 31 Mar 2017

Review comments on solid Earth Discuss10.5194-2017-13 The effects of tillage methods on soil aggregation 1 and crop yields in a 2 wheat-corn rotation under semi-arid conditions Dear Editor, This study has merit but I'm afraid this present document needs to be thoroughly reworked before publication is granted. There are many issues for this paper to reach the level of international publications but I feel this is feasible providing the authors dedicate enough effort on it. The first issue concerns the fact that any new research should convince on its novelty and this can only be done by (1) acknowledging the existing literature on the subject; (2) discussing the existing finding and identifying research gap(s); (3) clearly stating the research objectives. From the first few sentences of the abstract, it can easily be seen that the papers does not provide this kind

of information. The introduction section is also lacking presenting what has been done on the impact of tillage on grain yield and soil properties. The writing is not precise enough with main grammatical issues. The first sentence of the abstract below does not sound scientific: "wheat-corn two-course rotation system on the some soil aggregation properties and yields were investigated" what is "on the some soil aggregation properties" what type of "yields" is it about? Below are some tips With best regards

Abstract Abstract 14 In this study, the effects of different tillage methods under wheat-corn two-course rotation 15 system on the some soil aggregation properties and yields were investigated. Experiment 16 was laid out in a split plot design with three replications during four crop years. Subsoiler, 17 moldboard, sweep and chisel as main plots and rotary tiller and disc harrow as sub-plots 18 have been used in this study. The results showed that tillage methods were significant at 19 ($P<0.01$) as regards crop yields, and the highest yields as 6249 and 11720 kg/ha for wheat 20 and 9891 and 73080 kg/ha for corn grain and biomass were produced in subsoiler treatment, 21 respectively. Subsoiler+rotarytiller treatment was significant at ($P<0.05$) with 2.063 mm as 22 to mean weight diameter (MWD) value. The subsoiler and chisel were statistically in the 23 same group with regard of water stable aggregates (WSA) value, and it was significant at 24 ($P<0.05$) with 67,83%. Bulk density, total porosity and air porosity values were significant at ($P<0.01$), and 1.38 grcm-325 , 51.2% and 12.5% values were determined in rotary tiller 26 application, respectively. Field capacity (FC) and permanent wilting point (PWP) were 27 significant at ($P<0.05$) and ($P<0.01$) with 31.89% and 17.21% values in the chisel 28 treatment, respectively. Crop yields and positive effects on the physical properties were 29 considered subsoiler+rotary tiller treatment was the most successful, and it was followed by 30 chisel+rotary tiller treatment according to four-year study results.

Tips for scientific writing There are many different ways of writing an abstract and an Introduction. This depends on the academic subject involved, the journal itself and the specific topic of the article. It is important for the purpose of the research that

authors can identify the patterns used in abstracts of comparable articles published in the same area, and for journals that authors might write for. Abstract A. Topic sentence (s) on the subject (its importance) and research question(s): what is(are) the research gaps in this field of research? B. Objectives of the study C. Materials and methods used in the study D. Main results (with quantitative information, tests of significance) E. Conclusions: how these results respond to the objectives; general implications of the research

Introduction sections A. Presenting the background of the subject; B. Indicating the importance of the research on the subject; C. Acknowledging what has be done so far on the subject by referring to existing research studies and reporting ones; referring to methods and ideas associated with other researchers; D. Pointing to a gap in knowledge of the subject; E. Selecting research objectives F. Explaining the organisation of the research;

Discussion section may fulfil one or more of the following functions: A Presenting background information B Summarising what was (not) done C Explaining why it was (not) done D Evaluating the method(s) or model used E Statement of result(s) F Explanation of result(s) – why and how it happened G Implication of the result(s) – what it does, or does not, imply H Making reference to previous research I General statement of interpretation J Elaboration of interpretation K Discussing implication(s) of the interpretation L Rejection of interpretation M Acceptance of interpretation N Making a recommendation O Stating the limitations of the data P . . . . . . . . . . . . . . . . . . . . . . . . . . . . .. (other)

Conclusions A. Remind of research objectives B. Statements of general findings C. Statements of specific and significant finding D. Statement of overall trends with respect to what was known prior to the study E. How well do results respond to initial gaps, research questions F. Making predictions; recommendations.

---

## Author Comment (AC1) · 27 Apr 2017

**Revisionnotes**

"The effects of tillage methods on soil aggregation and crop yields in a wheat-corn rotation under semi-arid conditions" submitted by Hossein tabiehzad[1], Gokhan Cayci[2], Kiarash Afshar Pour Rezaeieh[3*] (MS No.: se-2017-13)

**1) Comments fromAnonymous Referee 1**

**General comment:**
The authors present a study about the influence of tillage methods on soil physical properties and crop yields in a wheat corn rotation in Iran which goes to subjects the journals publications scope. The paper is interesting, but it is not performed to be well understood. The paper have some interest in given subject, but shows several weaknesses which should be changed and discussed. Therefore I suggest reject &re-submission grade in order to improve quality.

Mayor points: The paper is "wordy", but on the other hand not focused in important findings (e.g. authors do not connect the yields and biomass with physical soils state at each treatment, neither explain their results). The reader gets lost in all the words.

Condense the manuscript. Remove all unnecessary statements and data. Please, avoid repetition and make sure that all statements are credible. The English is not understandable in many places. Also, there is lack of hypothesis in this study and novelty of the study can be explained. Effects of tillage management on plant, soil and environment certainly increase knowledge of how to build more sustainable agricultural systems in Iran. Nevertheless, this subject is extensively studied around the Earth so I wonder how attractive is this subject on international level? The main problem with the paper is that the authors did not provide description of treatments, tool that was used and depth of interventions. This makes the manuscript impossible to evaluate. Introduction section should be rewritten, while recent literature should be used to describe the actual problems. Paper can be better organised – I add some suggestions into text. Furthermore, your data deserves better discussion. Discussion is poor and does not explain neither supports the study results. The authors should say why they have it and not present only the results of the others. More details about sampling procedure are needed.

**Specific comments:**

Abstract should be rewritten. Please rewrite according next guidelines: Include more information to encourage readers to continue. Many only read title and Abstract. State where you perform the research. State objectives of research, and show whether data meet objectives. State hypotheses tested, and show whether data support hypotheses. Include international classification of soils.
Include more data to support statements, and give better summary of the manuscript. Line 14: define "two-course" Line 15: Define "some aggregation properties"? Is better to use "physical properties" or state each property. Line 16: delete "crop" Line 19: Is better to show the results in tonnes per ha. Please change in whole manuscript. Line 21-22. Unclear, please rewrite. Line 24-28. It is hardly understandable. Please ensure English improvement and editing from English speaker.
Introduction Please organize the introduction of the paper better. 1: General aspects of tillage impacts on the agro-ecosystems 2: Background about tillage impacts on soil properties that you are studying in this paper. 3: Impacts of tillage and implications trough soil properties (that you study) on crops that you study (generally and in similar agro-ecological conditions). 4: Justify why your study is needed and the novelty of your work. 5: Hypothesis that you want to test in your work and aim.
Line 38: write "agricultural" instead "production" Line 39: write "Soil and water conservation are important: " Line 43: Please, use one term, not many of them – e.g. soil organic carbon – soil organic matter Line 46-48: There is plenty of research availableto support this statement. Add newer literature.
Line 49-50: Avoid repetitions Line 50-51: add "distribution" below "rainfall" and "rainfall events"

after "intensity" Line 52: Please avoid generally writing. State which soil characteristics. Line 53-55. Statement is not true. It is not difficult to make intervention (subsoiling) but is expensive. Line 67: unify terms: zero tillage – no tillage – direct sowing. Line 67-69: State in which method? Line 71: Cultivation is tillage. Please delete "cultivation" Line 73-76: Avoid general statements. Instead is better to specify which parameter will be changed and how in short and long term. Support each mentioned parameter with literature source. Line 87: State the difference between excessive tillage and conventional tillage Line 88: write "conservation" instead "protected or reduced" End of introduction: state hypothesis and explain novelty. Introduction: Study that deals with comparison of subsoiling (ripping) with conventional tillage is missing and their effect on soil physical properties and yield.

Materials and Methods Better explanation of experimental design is needed. State the area of each treatment. State the area of sub-treatment. Explanation and description of tillage procedures, tools and depth of tillage is missing. Essential for paper. Line 102: Urumia or Urmia? Line 105: use "terrain" instead "field". Delete "with very low slope" Table 1: use mg/kg instead ppm Table 2: Check the temperature in June. Remove the comma in May Line 121: which year? Line 122: which year? Soil analyses: Sampling strategy is poorly written. How many disturbed samples per plot were collected? How many undisturbed samples per plot were collected? Same for subplots. How many disturbed and undisturbed samples in total? For undisturbed samples: from which depth? Or depths? Determination procedure for aggregate stability needs some more details: : :
Statistical analysis should be described well.

Results and discussion: This section need to be reordered: 1) bulk density, air filled porosity and hydraulic conductivity, 2) MWD and WSA, 3) field capacity and wilting point, 4) biomass and grain yields. Try to present results in order presented in tables and figures. Highlight most important finding, and state the significant differences between treatments rather than present relative numbers. Discussion is poorly written and does not support, neither explain given differences between treatments. Line 158 and 163: it is not true. Line 165: Deep tillage means nothing. Provide specific information about tillage operation. Line 165-166: compared to what treatment? Line 168: This statement is not supported with results Table 3: state moisture content in grain and biomass. Does numbers represent average of all four seasons? Provide main plot x subplot interaction. Same for each year. Line 173-196: Suggest to authors try to explain their results rather than report other findings. Find and compare your results with study results performed on similar textured soils and climatic conditions. Figures: remove horizontal lines. Unify x –axis and put white background. Table 4: unify digits, remove blank under "molboard" row Line 245: use g per cm3 not gr per cm3. Please make adjustment in whole manuscript. Table 5: change letters in bulk density Line 253: put space after 10 Line 257: Use term "vehicular" instead "farm" Line 258: define "fluffy" Line 258-260: please delete. Does not refers to present subject Line 261: define "effective pore" Line 263 and 265: put % behind numbers Line 293-296: Does this study deals with crop rotation effect? Please delete this Line 297-299: What is procedure with residues in this experiment? Add this information in M&M section Line 301-303: When you compare results with others please provide information about tool that were used and depth of intervention instead using terms "reduced" and "minimum". Otherwise is meaningless comparison.

Conclusions Line 349-350: Authors presents only single factor effect. If you want a draw a conclusion similar to written than you should test an interaction effect. This statement is not confirmed. Line 351-353: did you study subsoil layers also? Or just topsoil? Please delete this, it is not investigated. Make conclusions from your results. Line 366-369: Effect of subsoiling on silty clay soils do not last longer than season or two. Please adjust conclusion like: : : :.. "subsoiling is suggested in years when crops with demands for deeper rooting occurs in crop rotation."
Literature There are only two (2) sources in last 5 years. I suggest refreshing the paper with newer literature.

**2) Author's Response and Changesin manuscript for referee I**

**General Comment**

First of all, we thank to the referee to review the manuscript in a short time and read carefully.We are grateful for referee for sharing experience and knowledge with us. Referee critics are constructive and guiding for authors.

We would like to mention that authors attempted to remove unnecessary parts as much as possible and tried to make the article more readable in the direction of referee suggestions.

In this context;

Unnecessary repetitions were avoided.

Our hypotheses and novelty of the study was explained.

Detailed sampling procedure wasexplained.

Treatments related to soil tillage practices were explained in detail.

The relations between crop yields and soil properties were tried to be explained in more detail.

The article was triedbetter to organize by considering the referee critics.

Some parts of the manuscript were re-written and re-configured with referee suggestions. In this context, some explanations and references not directly related to the subject were removed and new explanations and references were added to new version.

**Specific comments**

Our answers to the critics and suggestions for each section of the article are presented below.

**Abstract**

This section was completely re-organized with referee suggestions.

"two-course" line 15 was deleted.

"crop" line 16 was deleted

"kg/ha" unit corrected as tonnes/ha (It was done in whole article)

Line 21-22 was rewritten as "Subsoiler+rotary tiller treatment gave the highest mean weight diameter (MWD) with a value of 2,063mm(P<0.05)".

Line 24-28 were rewritten as "When sub-plot means were considered bulk  density was the highest in disc harrow treatment  with a value of 1.39 grcm$^{-3}$ (P<0.05), total porosity and air porosity values were significant at (P<0.01) and the highest values were as 51.20% and 12.05% in rotary tiller application, respectively".

**Introduction**

Line 38 production changed as "agricultural"

Line 39 was corrected as"soil and water conservation"

Line 46-48 "Anonymous" was deleted a new literature (Kasper et al. 2009) was added.

Line 49-50 was deleted because ofprevious repetition.

Line 50-51 sentence was corrected as "Due to irregularities in distribution of rainfall with regard to time and intensity of rainfall events in arid and semi-arid regions"

Line 52 was corrected as "Moreover, excessive or incorrect tillage reduce aggregate wettability and decrease soil organic pool" and reference "Marinariet al. 2000" was removed.

Line 67 "direct sowing" was deleted.

Line65-67 was changed as 15 up to 30 percent of plant residues remain on the surface part of the soil at "conservation tillage" system

Line 71 "cultivation" was deleted.

Line 87 "excess tillage" was removed.

Hypothesis and novelty of the study were added end of introduction section.

**Materials and methods**

An explanation about treatment in relation to plot design was added to the 2.2.1 section as **"**Parcel sizes and space between blocks, main plots and sub-plots were given.

"Tillage depths were presented"

Soil sampling was explained in detail in a separate sub section as "soil collection" in material and method section as 2.2.2.

Explanation about sampling before version was moved 2.2.2 section and sampling method was rewritten.An explanation for undistributed soil sampling was added to the 2.2.2section

Line 102 was corrected as **"**Urmia".

Line 105 "with very low slope" was removed.

Table 1 "ppm" unit was changed as "mg/kg"

Temperature in June was corrected as "11.8" in Table 2.

Line 121 "comma in May was removed"

Line 121-122 experiment Trial years were specified.

Section 2.2.3 was re-written in detail in terms of statistical analyses.

**Results and discussion**

Results and discussion section was re-configured according to referee suggestion. In this context below mentioned sections constituted in results and discussion part

3.1 Grain and biomass yield of wheat and corn

3.2 Mean weight diameter of aggregates and water stable aggregates

3.3 Bulk density, air filled porosity and total porosity and hydraulic conductivity

3.4 Field capacity and wilting point

Line 158-163 placement error in Table 3 related to tillage methods was corrected.

Line 165-166 was corrected as "Arora (1991), reported that 30cm deep chiselling was positively effective reducing water stress in corn. On the other hand, Gajri et al. (1997) reported that sub-soiling with a single tine chisel down to 0.40 m depthpractices in semi-arid regions of India has been proved to be effective in utilization of water in sunflower production in semi-arid regions compared with conventional tillage." and these explanation transferred to a further segment in the same section.

Numbers in Table 3 represent average of all seasons.

Stated grain and biomass amounts were determined after harvesting for each year. Moisture contents in grain and biomass were not measured.

Horizontal linesin figures were removed and inserted white background. Blanks in Table 4 was corrected.

Line 245 and Table 5 "gr cm$^{-3}$" units were changed as "gcm$^{-3}$ "it was done also in whole manuscript.

Line 257 "vehicular" term was used instead of "farm".

Line 258-260 was deleted and reference was removed from reference list.

Line 261-261 was deleted reference was removed from reference list.

Line 265 "%" was put behind numbers.

Line 297-299 was deleted reference was removed from reference list.

**Conclusion**

Line 349-350 was rewritten according to referee's critics.

Line 351-354 was deleted.

Line 366-369 was rewritten according to referee's suggestions.

**Literature**

Newer references were added to the manuscript as much as possible

---

## Author Comment (AC2) · 27 Apr 2017

**Revisionnotes**

**"The effects of tillage methods on soil aggregation and crop yields in a wheat-corn rotation under semi-arid conditions" submitted by Hossein tabiehzad[1], Gokhan Cayci[2], Kiarash Afshar Pour Rezaeieh[3*] (MS No.: se-2017-13)**

**1) Comments fromAnonymous Referee II**

Review comments on solid Earth Discuss10.5194-2017-13 The effects of tillage methods on soil aggregation 1 and crop yields in a 2 wheat-corn rotation under semi-arid conditions Dear Editor, This study has merit but I'm afraid this present document needs to be thoroughly reworked before publication is granted. There are many issues for this paper to reach the level of international publications but I feel this is feasible providing the authors dedicate enough effort on it.

The first issue concerns the fact that any new research should convince on its novelty and this can only be done by (1) acknowledging the existing literature on the subject; (2) discussing the existing finding and identifying research gap(s); (3) clearly stating the research objectives. From the first few sentences of the abstract, it can easily be seen that the papers does not provide this kind of information. The introduction section is also lacking presenting what has been done on the impact of tillage on grain yield and soil properties.
The writing is not precise enough with main grammatical issues. The first sentence of the abstract below does not sound scientific: "wheat-corn two-course rotation system on the some soil aggregation properties and yields were investigated" what is "on the some soil aggregation properties" what type of "yields" is it about? Below are some tips With best regards Abstract 14 In this study, the effects of different tillage methods under wheatcorn two-course rotation 15 system on the some soil aggregation properties and yields were investigated. Experiment 16 was laid out in a split plot design with three replications during four crop years. Subsoiler, 17 moldboard, sweep and chisel as main plots and rotary tiller and disc harrow as sub-plots 18 have been used in this study. The results showed that tillage methods were significant at 19 ($P<0.01$) as regards crop yields, and the highest yields as 6249 and 11720 kg/ha for wheat 20 and 9891 and 73080 kg/ha for corn grain and biomass were produced in subsoiler treatment, 21 respectively.
Subsoiler+rotarytiller treatment was significant at ($P<0.05$) with 2.063 mm as 22 to mean weight diameter (MWD) value. The subsoiler and chisel were statistically in the 23 same group with regard of water stable aggregates (WSA) value, and it was significant at 24 ($P<0.05$) with 67,83%. Bulk density, total porosity and air porosity values were significant at ($P<0.01$), and 1.38 grcm-325 , 51.2% and 12.5% values were determined in rotary tiller 26 application, respectively. Field capacity (FC) and permanent wilting point (PWP) were 27 significant at ($P<0.05$) and ($P<0.01$) with 31.89% and 17.21% values in the chisel 28 treatment, respectively. Crop yields and positive effects on the physical properties were 29 considered subsoiler+rotary tiller treatment was the most successful, and it was followed by 30 chisel+rotary tiller treatment according to four-year study results.

Tips for scientific writing There are many different ways of writing an abstract and an Introduction. This depends on the academic subject involved, the journal itself and the specific topic of the article. It is important for the purpose of the research that authors can identify the patterns used in abstracts of comparable articles published in the same area, and for journals that authors might write for. Abstract A. Topic sentence (s) on the subject (its importance) and research question(s): what is(are) the research gaps in this field of research? B. Objectives of the study C. Materials and methods used in the study D. Main results (with quantitative information, tests of significance) E. Conclusions: how these results respond to the objectives; general implications of the research

Introduction sections A. Presenting the background of the subject; B. Indicating the importance of the research on the subject; C. Acknowledging what has be done so far on the subject by referring to existing research studies and reporting ones; referring to methods and ideas associated with other researchers; D. Pointing to a gap in knowledge of the subject; E. Selecting research objectives F. Explaining the organisation of the research;

Discussion section may fulfil one or more of the following functions: A Presenting background information B Summarising what was (not) done C Explaining why it was (not) done D Evaluating the method(s) or model used E Statement of result(s) F Explanation of result(s) – why and how it happened G Implication of the result(s) – what it does, or does not, imply H Making reference to previous research I General statement of interpretation J Elaboration of interpretation K Discussing implication(s) of the interpretation L Rejection of interpretation M Acceptance of interpretation N Making a recommendation O Stating the limitations of the data P : : :: : :: : :: : :: : :: : :: : :: : :: : :: : :: : :.. (other) Conclusions A. Remind of research objectives B. Statements of general findings C. Statements of specific and significant finding D. Statement of overall trends with respect to what was known prior to the study E. How well do results respond to initial gaps, research questions F. Making predictions; recommendations.

**2) Author's Response and Changes in manuscriptfor referee II**

**General Comment**

First of all, we are grateful for referee for sharing experience and knowledge with us.

We tried to make the article more readable in the direction of referee suggestions.

In this context;

Some parts of the manuscript were re-written and re-configured with referee suggestions.

We attempted to remove unnecessary parts in the manuscript as much as possible.

Unnecessary repetitions and references were avoided.

Our hypotheses and novelty of the study was explained.

Some explanations and references not directly related to the subject were removed and new

explanations and references were added and many rearrangements were done in new version

manuscriptby considering the referee critics.